# Maintenance Strategies for Industrial Multi-Stage Machines: The Study of a Thermoforming Machine

**DOI:** 10.3390/s21206809

**Published:** 2021-10-13

**Authors:** Francisco Javier Álvarez García, David Rodríguez Salgado

**Affiliations:** Department of Mechanical, Energy and Materials Engineering, University of Extremadura, Avda. Elvas s/n, 06006 Badajoz, Spain; drs@unex.es

**Keywords:** maintenance, sensors, multi-stage machine, maintenance algorithm, thermoforming

## Abstract

The study of reliability, availability and control of industrial manufacturing machines is a constant challenge in the industrial environment. This paper compares the results offered by several maintenance strategies for multi-stage industrial manufacturing machines by analysing a real case of a multi-stage thermoforming machine. Specifically, two strategies based on preventive maintenance, Preventive Programming Maintenance (PPM) and Improve Preventive Programming Maintenance (IPPM) are compared with two new strategies based on predictive maintenance, namely Algorithm Life Optimisation Programming (ALOP) and Digital Behaviour Twin (DBT). The condition of machine components can be assessed with the latter two proposals (ALOP and DBT) using sensors and algorithms, thus providing a warning value for early decision-making before unexpected faults occur. The study shows that the ALOP and DBT models detect unexpected failures early enough, while the PPM and IPPM strategies warn of scheduled component replacement at the end of their life cycle. The ALOP and DBT strategies algorithms can also be valid for managing the maintenance of other multi-stage industrial manufacturing machines. The authors consider that the combination of preventive and predictive maintenance strategies may be an ideal approach because operating conditions affect the mechanical, electrical, electronic and pneumatic components of multi-stage industrial manufacturing machines differently.

## 1. Introduction

The industrial production environment is becoming increasingly competitive, reliable and optimised. Industrial environments comprise several coordinated production lines and supplementary services that work towards achieving their production objectives.

Production processes are usually made up of several operation steps. Depending on the design of the production system, a common solution proposes using the same single-stage machines for each operation step. These days, there is another increasingly popular alternative based on multi-stage machines, in which the same machine carries out all the production phases.

From a maintenance viewpoint, in case of using single stage machines for different operation steps, any failure in one of the machines in a phase does not necessarily imply a production stoppage, although it may mean a temporary loss of the line’s production capacity. However, in industrial production systems based on multi-stage machines, a multi-stage machine is a machine that performs different consecutive operations within a production process. In this case, a failure in any machine component means a complete stoppage of the production line. As a result, the study of component reliability and availability is critical in this type of machine. 

Multi-stage machines are used in many industrial processes such as ultrasonic cleaning machines, terrine thermoforming machines, transfer solutions in packaging, fruit sorters, control solutions at logistic warehouse inputs and outputs.

Maintenance and availability monitoring strategies have evolved with time and changes in machine manufacturing technology. Preventive maintenance strategies are currently known to be the most popular [1]. In industrial machines, besides maintenance strategies based on predictive maintenance [2,3], statistical studies have also been carried out for prescriptive maintenance [3], conceptualisation based on Cyber-Physical Systems, artificial intelligence, Big Data [4] or even Digital Twin (DT) modelling [5].

### 1.1. Preventive Programming Maintenance (PPM)

This is the most popular maintenance strategy in the industrial environment. Taghipour, S. [6] studied this strategy by monitoring the degradation of components in production lines, using an exponential model to obtain the best maintenance strategy. Duffuaa, S. [7], however, related the study of PPM to monitoring and process decisions on a single-stage machine.

The study of the reliability of multi-stage machines provides interesting information for decision-making and PPM strategies. This strategy has already been used in studies by Panagiotis, H. [8] and Ahmadi, A. [9], which showed a model of machine reliability monitoring in which decisions on preventive or corrective maintenance were made based on observed reliability, although they did not consider the cost of maintenance. Zhen Hu [10] uses the health index to assess the remaining component lifetime on manufacturing lines.

David, J. [11] suggested PPM modelling based on knowledge of all the times involved in the repair and commissioning of the machine. Each component has its own Mean Time To Repair (MTTR) depending on its availability, installation difficulty and configuration (see Equation (1)). This analysis may reflect critical values that may affect the maintenance strategy for each component.

Liberopoulos, G. [12] analysed the reliability and availability of a process based on the reliability and availability of each component susceptible to failure or wear and tear.

### 1.2. Improvement Preventive Programming Maintenance (IPPM)

This is based on the PPM strategy. This maintenance strategy minimises component replacement times and increases component safety stock, resulting in a minimum MTTR value and increasing component availability. Gharbia, A. [13] analysed the relationship between stock cost and scheduled preventive maintenance time. This maintenance strategy is widely used on intensively operated multi-stage machines. A shutdown due to an unexpected failure entails high opportunity costs. IPPM is used for all components or for components with a high replenishment time.

### 1.3. Algorithm Life Optimisation Programming (ALOP) 

This is a proposed maintenance strategy that aims to improve the maintenance of the machines by making decisions based on analysing sensor signals and a predictive algorithm of the state of the most relevant components.

Knowledge of the wear and tear of components is a difficult task to model. Studies by A Molina and G Weichhart used information from specific sensors at strategic locations on machines or systems, which provided information related to production status, such as Desing S^3^-RF (sustainable, smart, sensing, reference framework) [14,15]. Decisions were made by computing the data obtained. As a complement, Molina, A. [16] developed the Sensing, Smart and Sustainable studies, where he introduced the environmental factor in the monitoring and managing of Cyber-Physical Systems (CPS).

Satish T S Bukkapatnam suggested the use of specific sensors for anomaly–fault detection in processes [17]. P Ponce proposed studies using sensors and artificial intelligence [18] for the agri-food industry. Ponce, P., Miranda, J. and Molina, A. [19] proposed using sensors, the interrelation of their measurements with the machine components and a data computation system as a strategy to learn about the real state of the machine components.

### 1.4. Digital Behaviour Twin (DBT)

Introducing Industry 4.0 in production processes paves the way for Smart Manufacturing [20,21] in the industry. In manufacturing multi-stage machines, DBT allows the study of new strategies based on collecting and processing data and defining standard behaviour patterns, which are then compared with real behaviours. This strategy provides essential information for decision-making based on the analysis of current behaviour and comparison of sensor readings.

Using smart devices, cloud computing [22], the study of Machine to Machine (M2M) strategies [23], while maintaining a high level of security and data quality based on international standards [24,25] is indispensable to achieve the full potential of Industry 4.0. Alsharif, M. and Rawat, D.B. [26] propose cloud-base service architecture form managing machine learning models that best fit different Internet of Things (IoT) device operational configurations for security. The necessary traceability in the value chain is possible with the application of the so-called Industry 4.0 [27,28,29].

Moreover, the conception of Cyber-Physical Systems (CPS) [30,31,32], Mixed-Criticality Systems (MCS) [33] or Industrial Cyber-Physical Systems (ICPS) [34] have prompted a change in the definition of systems, their monitoring and study to obtain the best information and interaction in real-time between a physical system and the monitoring, data computation, communication and interrelation with other systems [35]. K Meng’s paper, called Smart Recovery Decision-Making (SRDM) [36], uses data computation for end-of-life prediction of products.

Decision-making based on accumulated knowledge by the design and assessment of behavioural models is possible thanks to Behavioural Design Encapsulations defined by Stary, C. [37]. They are based on the reconfiguration of patterns with the accumulated knowledge of experience.

The emergence of the DT concept has made it possible to know and digitally simulate the behaviour of the physical model, and therefore improve control over the reliability and availability of equipment, as JdA Bertazzi [38] points out. However, for its application in multi-stage machines, a study and precise modeling of the physical behavior is required, in addition to subsequent adjustments and, finally, the verification that the model responds in the same way as the real model to external changes, boundary conditions or production [39,40,41,42].

Some studies define Evolutionary Digital Twin (EDT) as a parallel and complementary digital approach to DT and the real model [43]. Thus, knowledge of reality is also used as a source of learning for the system. This study allows the response of the model to be more flexible and adaptive to changes through supervised learning.

In a study by Wright, L. and Davison, S. [44], a DT is defined as an executable virtual model of a physical part or system. The digital model must then include the equations of the physical system and sensors that provide feedback on the real behaviour. Therefore, a DT can report on the correct or incorrect performance, decision-making or even prediction of the machine’s lifetime. The study also indicates that to achieve a behavioural model with DT, it must have sensors, be accurate in its calculations and be quick to suggest decisions.

Studies by Chakraborty, S. and Adhikari, S. [45] propose the modelling of a DT through the parallel study of response prediction and reality learning. A DT is used to simulate the behaviour of machines [46]. The study by Ritou, M. [47] defines the concept of “digital shadow” as a model that extracts information from the physical system, computes the values and proposes decisions on the state of the machines.

Few references dedicated to maintenance management in industrial manufacturing multi-stage machines have been found during the search for references.

### 1.5. Methodology of the Case Studied

This paper, however, studies a real case of a multi-stage thermoforming machine with a capacity of six terrines per cycle and a cycle time of 4 s. Four maintenance strategies were studied for one year: two usual preventive maintenance strategies (PPM and IPPM) and two predictive maintenance proposals (ALOP and DBT), adapted to Multi-Stage Machines technology.

The work carried out in this research is based on the use of four different maintenance strategies whose operation was observed for one year. The thermoforming multi-stage machine was working continuously 8 h a day, Monday to Friday, for a year. To carry out the work, the following steps were followed in order:Conceptualisation of the machine. In this section, the most important components, whose reliability, efficiency and availability were to be studied, were selected;Analysis of the causes and consequences of a failure in the selected components. (See Table 1);Proposition of individual maintenance times per component, as well as equations for calculating reliability, efficiency and availability;Proposition and location of appropriate sensors whose values are associated with the proper functioning of the components;Proposition and development of algorithms for ALOP and DBT strategies;Location of a master linear axis for the case of DBT, by means of which the study is related to the position of the encoder and subsequently converted to units of time;Configuration of the Programmable Logic Controller (PLC) datalogger function and record all the relevant values in each strategy;Recording of the failures and errors detected in ALOP and DBT;Evaluation of the results obtained.

The objectives proposed in this study are:Obtain a systematic approach to managing the maintenance of multi-stage machines, so that it can allow their use not only in the case studied;Evaluate and compare the results that are obtained with the different of maintenance strategies;Propose a maintenance strategy for the detection of unexpected failures that cause manufacturing without expected quality or production stoppage.

## 2. Case Studied

Production in small packages, known as single use, is increasingly present in the industrial environment. Commonly used products such as oil, vinegar, etc., are already marketed on a large scale by many industries that produce them in large production batches. Figure 1 shows an image of the multi-stage thermoforming machine studied in this article.

This multi-stage thermoforming machine consists of:A structural, fixed part, usually not subject to wear and tear but must be adequately protected against corrosion and meet health and food standards;Electronic components, power actuators, servo drives, motors, gearboxes, variable speed drives, electrical and electronic devices, including the HMI operator terminal, which are usually 4.3, 7 and 10 inch touch screens;Mechanical components subject to movement, such as bearings, shafts, belts and cams. They are generally designed with fatigue-resistant materials but may be damaged by wear and tear and environmental conditions;The peristaltic and pneumatic drive system, with which the filling of the terrines and the upward and downward movements of sets of cylinders for adhesion, sealing, glueing and cutting of the terrines are produced, respectively. These systems have bronze bushings, which often suffer from wear and tear;A polymer roll dosing system for the top and bottom of the tray. The movement of these rollers is carried out as required at any given moment.

Improvements in process monitoring and technology have made this type of machine controllable by PLC that receive status signals from the field and act on the power actuators for the coordinated execution of all movements. The same technology can be used to manage the availability of the machine or its components.

Table 1 shows a basic decomposition of the components of the machine subject to failure in this paper. A distinction is made between static or moving elements, the possible fault source and the consequence of its failure.

Multi-stage thermoforming machines are one of many multi-stage machines in industrial manufacturing processes. These machines comprise several sub-processes ranging from the management of the polymer film to the container and lid, including the dosage and final cut. Figure 2 shows the steps of this machine ordered sequentially.

Production capacity can vary from 6 to 12 terrines in the last step, depending on whether the machine is designed for manufacturing 3, 6, 9 or up to 12 tubs simultaneously. Normally, production is carried out with thermoforming moulds of 2, 4, 6 and 12 tubs, composed of one or two rows according to the design of the multi-stage thermoforming machine, then in one cycle, up to 12 tubs can be manufactured simultaneously. This affects the size, the mould of the thermoformer, the number of peristaltic pumps, the rails for the row passage, the lid’s thermal bonder and the tub cutter’s size. Here, the thermoforming mould used is for six tubs, and the cycle time is 4 s. 

Figure 3 shows the terrine used. It is possible to see the lid and the tub. When the lid is added by Step six, terrine is obtained.

Standard operation requires the constant coordination of all sub-processes since a failure in one of them means production stoppage. There is a master linear axis (see Figure 1) in the lower part of the machine that runs from the thermal conditioner of the polymer for the thermoformer container to the cutter for finished tubs, which permits coordinated movements with cams in synchronised positions to ensure the process is controlled at a constant speed.

It can be understood that a critical component failure can lead to a failure of the whole machine either because it works without the necessary quality or because it cannot continue with the commissioned work.

The times involved in the study of the failures [11,12], are:TTRP: Time to replace a component;TTC: Time to configure;TTMA: Time to mechanical adjustment;TTPR: Time to provisioning;MTTR: Mean time to repair;MTTF: Mean time to failure;MTBF: Mean time between failure;TTLR: Line restart time, defined by expert knowledge;TLP: Time lost production.
(1)MTTR= TTRP+ TTC+TTMA+ TTPR
(2)TLP=MTTR+ TTLR
(3)MTBF=MTTR+MTTF
with these times, two concepts are used: efficiency (4) and availability (5). Both concepts will be used as indicators of success in the preventive control of machine failures.
(4)$$\mathrm{Efficiency}=1-\frac{\mathrm{TLP}}{\mathrm{MTTR}+\mathrm{MTTF}}$$
(5)$$\mathrm{Availability}=\frac{\mathrm{MTBF}}{\mathrm{MTBF}+\mathrm{MTTR}}$$

## 3. Maintenance Strategies for the Multi-Stage Thermoforming Machine

The maintenances assessed in an initial phase on this multi-stage thermoforming machine have been PPM and IPPM. High levels of availability and efficiency are achieved. ALOP and DBT strategies have been assessed, and failures were detected before the static value of MTTF (see Table 2) determined by PPM and IPPM.

### 3.1. PPM: Preventive Programming Maintenance

This strategy is based on using existing data from the usage of the machine. With the information gained from the usage of the machine, each component has its own time values (TTRP, TTC, TTMA, TTPR, MTTR, MTTF, MTBF, TTLP, TLP), and an individual value for availability and efficiency.

The results of setting the line restart time, TTLR, at 14.400 s and using stable market values (values obtained from manufacturers and experience) for the times in this machine are shown in Table 2:

Using the exponential function given by expression 6, the reliability of all the components is calculated in a time equal to MTTF. Figure 4 shows the results.
(6)$$R_{\left(t\right)}={\;e}^{-\lambda t}$$
where λ factor is the inverse value of MTBF [48] if we consider the constant fatigue of components.

### 3.2. IPPM: Improvement Preventive Programming Maintenance

Table 2 shows the TTPR value for all items. It is a significant value when calculating the MTTR value (see Equation (2)).

The IPPM strategy is based on the TTPR of components that would considerably reduce the value of MTTR, and consequently, in the efficiency and availability values. Table 3 shows the results of substituting the TTPR for a residual search time in own stock. Then if the component fails and needs to be replaced, the TTPR value affects the MTTR very little and therefore increases the availability and efficiency of the machine (see Equations (4) and (5)).

The results obtained reveal very high efficiency and availability values for PPM (see Table 2) and IPPM. Components with a high TTPR value improve their efficiency and availability values. Comparison of the results between the two provides a maximum increase in efficiency in 9.26% and availability by 8.4%. Figure 5 shows a comparison of these results. 

In other components such as 2, 3, 10, 11, 14, 15, 21 and 22 there has also been an increase in efficiency and availability above 3%.

The results obtained reveal that availability and efficiency improve with the implementation of the IPPM strategy.

The results show that electronic components such as the PLC, HMI, temperature controller, solid state relay, pressure sensor, servo drive form peristaltic pump, peristaltic pump and absolute encoder improve their availability with this strategy, while mechanical components such as the bronze cap, linear axis, linear bearing, pneumatic valve, pneumatic cylinder and terrine cutter partially improve their availability. Consideration of market conditions, transport problems, supply problems or health scares can increase the value of TTPR. These events do not affect the IPPM strategy because it is based on having the components in stock. To avoid affecting the PPM strategy, the TTPR value should be changed by frequently consulting the market for this time in all components. The availability and efficiency of the machine can be maintained in this case and do not decrease due to external causes if a failure occurs.

### 3.3. ALOP: Algorithm Life Optimisation Programming

The MTTF of each component can be changed with this strategy by analysing the behaviour of measurements from various sensors. This strategy would enable optimising the useful life of each component. This strategy is compatible with maintenance decisions, and conclusions of the previous strategy can be applied by the algorithm.

Figure 6 shows this model, in which the PLC that manages the process is the same equipment that manages the ALOP algorithm. It consists of sensors in specific parts of the multi-stage thermoforming machine. The real-time processing of the values measured by the sensors allows to know the status of the components and calculate the MTTF in real time. This quality allows a failure to be detected before it occurs. Compared to the PPM and IPPM strategies that keep the MTTF at a fixed value, this strategy detects failures before a static time (remember static MTTF in the PPM and IPPM strategies). The possibility of detecting failures before the fixed MTTF value proposed in PPM or IPPM causes the lower efficiency and availability values of this strategy compared to the two previous strategies (see Equations (4) and (5)).

Table 4 contains the sensors used in the multi-stage thermoforming machine and the component group they affect.

All sensors provide an analogue output signal. A datalogger oversees monitoring, recording and treating the signals in real-time.

#### Mathematical Model of the Algorithm

The adoption of this model is based on the accumulated experience in the usage of the PPM and IPPM strategies in the multi-stage thermoforming machine. ALOP was implemented when specific components with available lifetimes according to their proposed MTTF in PPM or IPPM were experiencing unexpected failures. Poor knowledge of the causes of such failures and the impossibility of solving this problem with PPM or IPPM led to the creation of ALOP in an attempt to correct the MTTF value according to the reality measured by sensors reporting to the process control PLC.

This algorithm proposes the calculation of reliability parameters such as MTTF by using the values of distributed sensors that provide information on physical magnitudes whose normality values are recorded. The aim is to compare and adjust the times before failure to then adjust the MTTF value for each component and calculate the component’s reliability using the exponential model. As a complement to the algorithm, a warning factor (WF) indicating an unacceptable value of a sensor will be proposed.

The application of this ALOP model focuses on components not kept in stock that cause machine downtime and whose failure causes a considerable TLP value (see Equation (2)). Components such as command and signalling (buttons, switches), a master power switch, plug-in relay and safety components do not apply to this model due to being components of very low cost and high availability of stock.

Equations (7) and (8) are proposed for the calculation of MMTF_i_(t). A step-by-step algorithm will then be proposed to enable decision-making:(7)$${\mathrm{MTTF}}_{i}\left(t\right)=\left[{\mathrm{MTBF}}_{i,0}-\left(t-t_{0}\right)\right]{\mathrm{fc}}_{\left(i\right)}-{\mathrm{MTTR}}_{i}$$
where MTBF_i_ is the mean time between failures of component “i”. This value is shown in Table 2, which results from adding the MTTF and MTTR values for each component proposed in the PPM and IPPM strategies. MTTR_i_ is the mean time to repair a failure of equipment “i”. ${\mathrm{fc}}_{\left(i\right)}$ is a correction factor for component “i” that depends on the measurements of its associated sensors and is calculated every 100 machine cycles (Since the cycle time is 4 s (see the beginning of Section 2) and therefore 100 cycles correspond to 400 s, it is considered a reasonable time to take measurements on the sensors) and corresponds to the following equation:(8)$${\mathrm{fc}}_{\left(i\right)}=\overset{n}{\underset{j=1}{\prod }}\frac{\sigma_{\left(t\right)j,i}}{\sigma_{\left(t+100\right)j,i}}$$
where $\sigma_{\left(t\right)i,j}$ is the standard deviation at time “t” of the measurement of sensor “j” whose evolution can provide information on the reliability and availability status of component “i”. $\sigma_{\left(t+100\right)j,i}$ is the standard deviation at time “t + 100” of the measurement of sensor “j”, the evolution of which can provide information on the reliability and availability status of component “i”.

The risk function described in D M Frangopol’s study [49] is then used for each component:(9)$${\mathrm{fr}}_{\left(t,i\right)}=(1-R_{\left(t,i\right)}){\;\mathrm{Cf}}_{i}$$
where ${\mathrm{fr}}_{\left(t,i\right)}$ is the risk in economic terms based on the reliability of component “i” at time “t” and R_(t,i)_ is the reliability of component “i” at time “t”, which is calculated using the exponential model $R_{\left(t,i\right)}={\;e}^{\frac{-\lambda }{t}}$, where λ coincides with 1MTBFi−LC where ${\mathrm{MTBF}}_{i-\mathrm{LC}}$ is the mean time between failures of the previous assessment time of component “i”. ${\mathrm{Cf}}_{i}$ is considered constant and is the cost in economic terms of the TLP due to a failure to be repaired in component “i”.

The risk factor ${\mathrm{fr}}_{\left(t,i\right)}\;$ is used to advance sourcing decisions for component “i” even if the algorithm has not yet suggested it. It is essential to define risk margins for each component so the value of ${\mathrm{fr}}_{\left(t,i\right)}$ must be within the margins set by the user. The lower the reliability of a component $R_{\left(t,i\right)}$, the higher its failure function $F_{\left(t,i\right)}=1-{\;R}_{\left(t,i\right)}$. Therefore, the product between $F_{\left(t,i\right)}$ and the constant value ${\mathrm{Cf}}_{i}$ will become larger and larger until it reaches ${\mathrm{Cf}}_{i}$($R_{\left(t,i\right)}=0$). Here, the component fails, and the value of ${\mathrm{fr}}_{\left(t,i\right)}$ is maximum (see Equation (9)). The comparison between ${\mathrm{fr}}_{\left(t,i\right)}$ is used as an indicator for the acquisition of component “i”.

The warning level or technical alarm WF is an inadmissible value for each sensor, set as a technical warning threshold indicating which components may be affected by the warning. This warning may lead to a decision to procure the component or replace it if it is in stock. A Gaussian distribution criterion based on the confidence level of the sample of values is used for verification. The following equation is used:(10)$$\mathrm{WF}>\bar{{\mathrm{SA}}_{j}}\pm {\;c}_{i}\times {\;\sigma }_{j}$$
where $c_{i}$ expresses the confidence level or permissiveness of accepting or not accepting deviations from the mean measured value of each sensor. Following the Gaussian Normal distribution criterion, the smallest value of “c” is 0.67 [50], corresponding to a confidence level of 50% of the measured values. Each sensor can have a different value of c_i_ depending of the dispersion of its measurements. In this study, $c_{i}=0.67$ was used for all “j” sensors, because it is a restrictive criterion in the Gaussian distribution, so that the algorithm will be more sensitive to variations that are far from the mean value of the sensor measurement.


**Proposed ALOP algorithm:**
**STEP 1.** The time for evaluation and recalculation of values is set as t = 1000 s.**STEP 2.** From t = 0, values are taken from the “j” sensors measurements, SA_j_. every 10 s.**STEP 3.** $\bar{{\mathrm{SA}}_{j}}$ And σ_j_ is calculated every 100 s.**STEP 4.** At t = 1000 s, ${\mathrm{fc}}_{\left(i\right)}$ is calculated for each component “i”.**STEP 5.** The values ${\mathrm{MTBF}}_{i,1000}$ and ${\mathrm{MTTF}}_{i,1000},\;R_{i,1000},{\;\mathrm{Eff}}_{i,1000},{\;\mathrm{AV}}_{i,1000}$ are calculated.**STEP 6.** The value of ${\mathrm{MTTF}}_{i,1000}$ is compared with ${\mathrm{MTTR}}_{i}$ and subsequently with ${\mathrm{TTPR}}_{i}$.**STEP 7.** The risk factor ${\mathrm{fr}}_{\left(1000,i\right)}$ of component “i” is calculated. It is compared to the cost of component “i”:(11)$$\mathrm{IF}\;{\mathrm{fr}}_{\left(1000,i\right)}>{\mathrm{Component}\;\cos t}_{i}\to \mathrm{Component}\;\mathrm{supply}\;\prime \prime I\prime \prime \;\to \;\mathrm{Component}\;\mathrm{supply}\;\prime \prime I\prime \prime$$
(12)$$\mathrm{IF}\;{\mathrm{fr}}_{\left(1000,i\right)}<{\mathrm{Component}\;\cos t}_{i}\;\to \mathrm{No}\;\mathrm{decisions}$$**STEP 8.** If ${\mathrm{MTTF}}_{i,1000}<{\;\mathrm{MTTR}}_{i}\;,$ the notification for acquiring component “i” is initiated.**STEP 9.** If there are no warnings in Steps 7 and 8, compliance with the following is verified:(13)$$\mathrm{IF}\;{\mathrm{WF}}_{j}<\bar{{\mathrm{SA}}_{j,1000}}\pm {\;c\;x\;\sigma }_{j,1000}\;\to \mathrm{No}\;\mathrm{decisions}$$
(14)$$\mathrm{IF}\;{\mathrm{WF}}_{j}>\bar{{\mathrm{SA}}_{j,1000}}\pm {\;c\;x\;\sigma }_{j,1000}\;\to \mathrm{Technical}\;\mathrm{warning}$$**STEP 10.** At t = 1000 s, MTBF_0_ values are updated to MTBF_1000_ since the 1000 s that has elapsed is to be deducted from the mean time to failure of component “j”.**STEP 11.** Start the algorithm again at Step 2.


This algorithm was adjusted successively over 1 year. In the conclusions, the results of ALOP will be compared with DBT and the effectiveness of their respective algorithms.

### 3.4. DBT: Digital Behaviour Twin

This strategy proposes using a real-time model that maps the outputs to actuators of the process control (PLC). The monitoring of these variables reports the real operating status of the machine in the order to know which commands are being executed, which field signals are being measured and their values. This strategy uses the position of the absolute encoder, which measures the position of the main shaft of this multi-stage machine. Depending on the position in each cycle, the commands representing the expected behaviour of the process are activated in a coordinated order.

Figure 7 shows the schematic of the DBT model setup for this strategy. It uses the same sensors as ALOP (see Table 4). In this strategy, the activations and deactivations of the actuators are monitored, and the sensor values and the position of the absolute encoder are compared with a so-called normal behaviour pattern. An essential difference to the ALOP strategy is the use of a different measurement scale. ALOP assesses the sensor measurements according to the time algorithm, whereas DBT uses the assessment of the sensor measurements in terms of the position taken by the absolute encoder (see item 25, Table 2).

The DBT strategy proved to be efficient in this paper and can, therefore, be considered appropriate for developing maintenance strategies for other industrial multi-stage machines. The position of the main shaft of the machine is known through the encoder. The decision-making provided by the proposed DBT algorithm is performed on the time scale by converting the encoder position to time.

In this machine, a work cycle starts at position 0 and ends at position 999 of the absolute encoder. All sensor measurements are linked to machine actuators. They are then recorded and stored according to the encoder position. As a result, a behavioural pattern is obtained with sensor measurement values within the maximum and minimum thresholds and is considered the standard behavioural reference for the multi-stage thermoforming machine. During the normal operation of the machine, the real values are compared with the standard to determine whether the machine is working correctly. The strategy also studies the trend of sensor values and whether they show a potential risk to component lifetime or manufacturing quality.

If the encoder position indicates it and a “z” Actuator (AC_Z_) is activated, this input is represented with value one or zero if not activated. The SA_i_ sensors in Table 4 provide measurements throughout the cycle regardless of activations or non-activations of the AC_Z_ actuators. All SA_i_ sensors have a nominal, minimum and maximum value. The decision to assess or replace the component is made based on the analysis of the measurement trend of its associated SA_i_ sensors and the maximum and minimum values allowed for these measurements.

Table 5 shows the pattern of behaviour of the machine from encoder position 0 to 999. The study has evaluated both the state of the actuators and the value of the sensors every 10 incremental positions of the encoder.

The relevance of sensor measurements can be recognised by using the encoder position.

Therefore, this strategy allows maintenance to be managed by adjusting the operating time of components at the end of their useful life or when they may be damaged by external causes and need to be replaced.

#### DBT Mathematical Model

Since the normal behaviour pattern and the nominal, maximum and minimum values of the sensors at all encoder positions are known, Artificial Intelligence procedures are not necessary. This feature is considered an advantage of this strategy.


**Proposed DBT algorithm:**
**STEP 1.** The assessment procedure starts every 10 encoder positions (EP_10_ to EP_1000_).**STEP 2.** An assessment is carried out every 10 positions:
Actuator values AC_z_ (binary value zero or one);Values of SA_i_ sensors (analogue signals)
**STEP 3.** Pattern checks:
The AC_z_ activations reading for the Encoder Position (EP) 10 value should coincide with the valid pattern (see Table 5)If not → PLC or encoder fault.The SA_i_ sensor reading for the EP10 value should coincide with the valid standard (see Table 5)If not → Step 4.The AC_z_ activations reading for the SA_i_ value should coincide with the valid pattern (see Table 6)If not → Step 4.
**STEP 4.** Checking deviations of SA_i_ sensors:(15)$$\mathrm{If}\;{\mathrm{SA}}_{i,\;}\;\in \;\left({\mathrm{SA}}_{i,\mathrm{VN}}-\left|d_{\min}\right|,{\;\mathrm{SA}}_{i,\mathrm{VN}}+\left|d_{\max}\right|\right)\;\to \;\mathrm{No}\;\mathrm{decisions}.\;$$
(16)$$\mathrm{If}\;{\mathrm{SA}}_{i,\;}\;\notin \;\left({\mathrm{SA}}_{i,\mathrm{VN}}-\left|d_{\min}\right|,{\;\mathrm{SA}}_{i,\mathrm{VN}}+\left|d_{\max}\right|\right)\;\to \;\mathrm{Assessment}\;\mathrm{of}\;\mathrm{components}\;\mathrm{associated}\;\mathrm{with}\;{\mathrm{SA}}_{i}\;\mathrm{sensor}\;\to \;\mathrm{Step}\;5$$ where d_max_ and d_min_ are the maximum and minimum deviations allowed in the measurements of the “i” sensors.
**STEP 5.** The trend is assessed by analysing the mean and standard deviation of the last 1000 cumulative measurements of the SA_i_ value of sensor “i”, whose value is other than zero.

(17)
$${\bar{\mathrm{SA}}}_{i\left(10-1000\right)}=\sum_{\mathrm{EP}=10}^{\mathrm{EP}=1000}\frac{{\mathrm{SA}}_{i-\mathrm{EP}}}{1000}$$



For this calculation, the SA_i_ values that must have a defined value is different from than zero according to the behavioural pattern will be considered.
(18)$$\sigma_{{\mathrm{SA}}_{i}}^{\mathrm{EP}=1000}=\frac{\sum_{10}^{1000}{\left({\mathrm{SA}}_{i-\mathrm{EP}}-{\bar{\mathrm{SA}}}_{i\;\left(10-1000\right)}\right)}^{2}}{1000-1}\;$$

Based on the above values and assuming a Gaussian probability distribution, it is evaluated if the value is included in a statistical limit based on the previous measures.
(19)$$\left({\mathrm{SA}}_{i,\mathrm{VN}}-\left|d_{\min}\right|,{\;\mathrm{SA}}_{i,\mathrm{VN}}+\left|d_{\max}\right|\right)\in \;\left(\;{\bar{\mathrm{SA}}}_{i\left(10-1000\right)}\;\pm 3{\;\times \;\sigma }_{{\mathrm{SA}}_{i}}^{\mathrm{EP}=1000}\right)$$

If the trend is maintained, the algorithm calculates the time remaining before the SA_i_ sensor measurement can indicate a failure and/or an undesired shutdown.

The result is studied on the encoder scale. Therefore, the result is obtained in the number of cycles missing for the measurement of a sensor to go beyond its limits. NCTF_SAi_ is the number of cycles to failure indicated by the SA_i_ sensor.

**STEP 6.** Decision taking.

Once the study of the trend of the SA_i_ sensor values in the encoder position has been completed, the relationship between the encoder position scale and the time is defined. In this case (see beginning of Section 2):1 work cycle ≡1.000 encoder positions from 0 to 999=4 s
(20)$${\mathrm{MTTF}}_{\mathrm{SAi}}={\;\mathrm{NCTF}}_{\mathrm{SAi}}\times \;\mathrm{TC}\;$$

Expression (20) can calculate the number of cycles that can be performed with the sensor values within their maximum and minimum thresholds.

The DBT strategy and the encoder assessment scale in the maintenance management of the multi-stage thermoforming machine makes it possible to ascertain:Deviations in the measurements of the “j” sensors, whose relationship is established with the “i” items by Table 4;Whether the evolution of any of the “z” actuator activation/deactivation commands is correctly coordinated and is proceeding according to the normal pattern;If any of the measurements of the “j” sensors conform to the encoder position;If any of the measurements of the “j” sensors conform to the activation pattern of the “z” actuators at each encoder position;Whether the absolute encoder is providing the shaft position information correctly;Whether the process control, PLC, is executing the commands correctly according to the encoder position.

It also makes it possible to:Take early decisions on machine components and prevent unwanted faults by assessing the measurements of each sensor and observing the measurement trend;Know the planned production that can be performed without a failure;Adjust the d_max_ and d_min_ values for each SA_i_ sensor, allowing the establishment of a confidence margin where the output meets industry quality standards;Very precise control of deviations from nominal measurements of the “j” sensors by being assessed only when indicated by the position of the encoder and the “z” actuators and the sensor shows a value other than zero (see Step 5 of the DBT algorithm).

## 4. Results and Conclusions

The ALOP and DBT strategies have been tested on the multi-stage thermoforming machine working continuously 8 h a day, Monday to Friday, for a year. Table 6 shows the number of unexpected failures, with information on the warnings of each algorithm and which have warned of a real failure, and which have not.

Unexpected failures can be detected with ALOP and DBT algorithms. However, the ALOP algorithm has shown false warnings. The authors consider this may be due to ALOP taking measurements from each sensor every 10 s, whereby the nominal measurement value of the sensor or zero value may be recorded. As a result, the dispersion of measurements may be excessive. Increasing this dispersion may cause false warnings (see expression 8). For the DBT model, the trend of the measurements is only assessed on the measured value, which will always be very close to the nominal value unless the sensor fails.

As a follow-up, the DBT algorithm has detected unexpected failures in mechanical items 16 and 18. Failures in affected components are detected if the deviations in the SA_6_ and SA_7_ sensors are greater than 0.5 mm. The detection of possible failures in mobile mechanical equipment requires a maintenance strategy in which the assessment of deviations is as accurate as possible, with DBT being the best alternative.

Item 25 (encoder) suffered an accidental mechanical shock. From that moment on, its operation was not correct as the commands executed to the actuators started to be carried out without the expected coordination. Step 3 of the DBT algorithm warned very quickly, in less than one cycle. ALOP did not detect it because it uses the SA_1_, SA_2_ and SA_3_ sensors for that component, and none of the three sensors noticed an anomaly in the measurements. As a consequence, the machine was stopped by an operator.

As both algorithms detected failures in some components, the MTTF was reduced. To manage the maintenance of this alteration, the MTTF value of components triggered component replacement decisions as the mean time to failure was reduced and, therefore, the component’s lifetime ended. Their Efficiency and Availability values changed (see Equations (4) and (5)).

Figure 8 and Figure 9 show the comparison of Efficiency and Availability in percentage values of the components that presented unexpected failures detected by ALOP and DBT, and their values obtained in PPM and IPPM (see Table 3).

The detection of failures before the MTTF stated in the PPM and IPPM strategies is the consequence of the decrease in the efficiency and availability values of the affected components. However, the relevance of the decrease in the values can be compared to the advantages of detecting a failure before an unexpected stoppage and the opportunity costs it may entail (proposed for future research).

The study of maintenance strategies for multi-stage machines can be an avenue for future research. Through the results obtained, a solution is offered for unexpected failure detection, which for this type of machine is of great importance. With the results obtained, these conclusions can be drawn:The algorithms proposed for the ALOP and DBT strategies show favorable results, and their use can be proposed for managing the maintenance of other multi-stage machines;Because multi-stage machines require better maintenance control to detect unexpected failures, ALOP and DBT can be proposed as suitable strategies for this type of machine;Unexpected failures can be detected with ALOP and DBT strategies. The authors consider that both strategies complement PPM or IPPM, and their combined study could be an avenue for future research;The accuracy of the measurement evaluation procedure of the DBT strategy allows the detection of faults in moving mechanical components with very low deviations from nominal values;Knowledge of a normal operating pattern of machines is a very reliable source of knowledge for maintenance management. It allows the best assessment of component lifetime by setting limit deviations (d_max_ and d_min_) (See Step 4 in the DBT algorithm) on sensor-measured values, based on the quality standards of each industry;The detection of unexpected mechanical or electronic components failures may be due to alterations of environmental operating conditions and non-recommended voltage values;The knowledge of the production that can be performed without failures is only achieved with the DBT model;The IPPM application offers improvements of efficiency and availability and minimises MTTR, but stock costs can grow;Improvements in the efficiency and availability of the electronic components (see components 2, 3, 4, 10, 11, 14, 15, 21 and 22 in Figure 5) and partially the mechanical components (see components 10, 16, 17, 18, 19 and 23 in Figure 5) are noticeable. As with PPM, this strategy also fails to detect unexpected failures;Applying PM techniques based on the time scale is interesting if the SA_i_ sensor values provide constant and similar measurements throughout the process. Otherwise, the dispersion in values may not correctly reflect reality. On multi-stage thermoforming machines, it is very beneficial to evaluate the measurements on the scale of the encoder positions and then decide on the time scale.

The authors consider the following avenues for future research:Comparative study between the decrease in efficiency and availability by applying ALOP and DBT strategies, and the benefits of detecting unexpected failures compared with static value of MMTF provided by the PPM and IPPM strategies;Study of the application of different maintenance strategies for each kind of component in the same multi-stage machine;Study of the cost of the different maintenance strategies in a multi-stage machine.

## Figures and Tables

**Figure 1 sensors-21-06809-f001:**
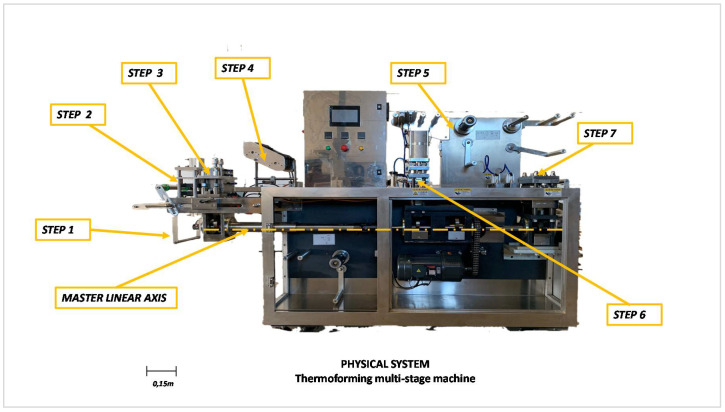
A thermoforming multi-stage machine of 6 terrines per cycle.

**Figure 2 sensors-21-06809-f002:**
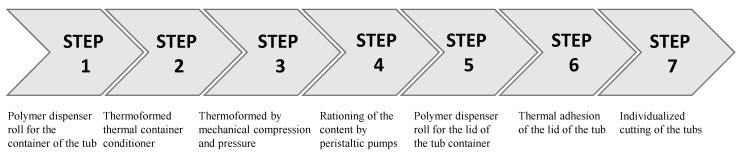
Sub-process in thermoforming and terrine filling machines.

**Figure 3 sensors-21-06809-f003:**
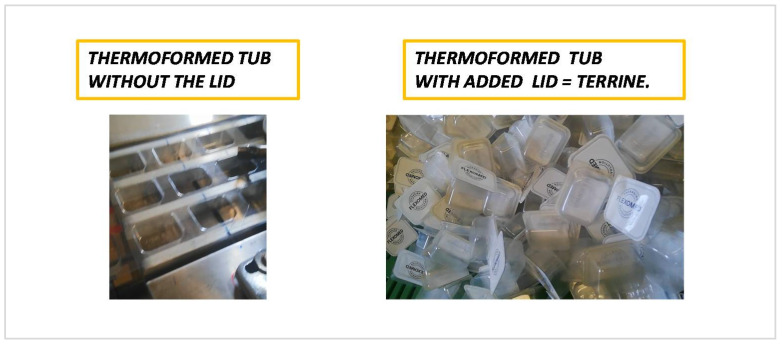
Example of terrine obtained in the thermoforming multi-stage machine studied.

**Figure 4 sensors-21-06809-f004:**
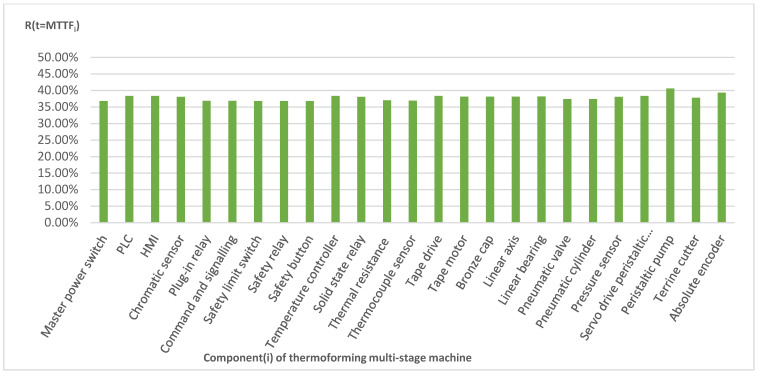
Reliability calculated at MTTF with the exponential function (6) in the PPM strategy.

**Figure 5 sensors-21-06809-f005:**
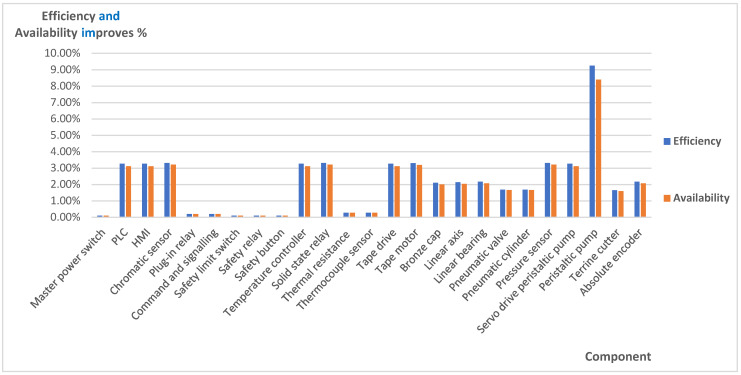
Percentage of improvement in efficiency and availability using IPPM strategy in terms of PPM.

**Figure 6 sensors-21-06809-f006:**
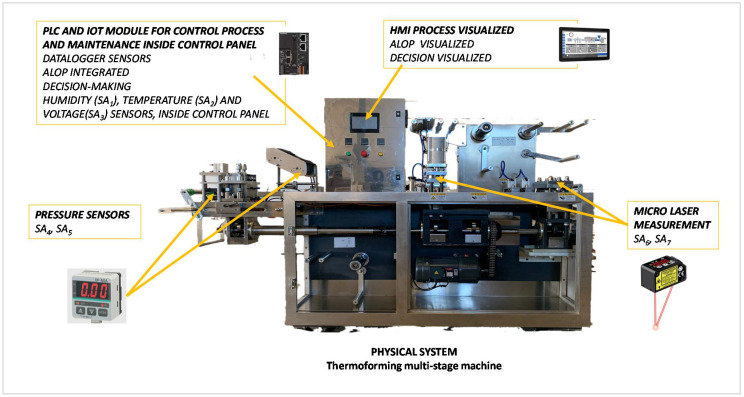
The setup of ALOP strategy.

**Figure 7 sensors-21-06809-f007:**
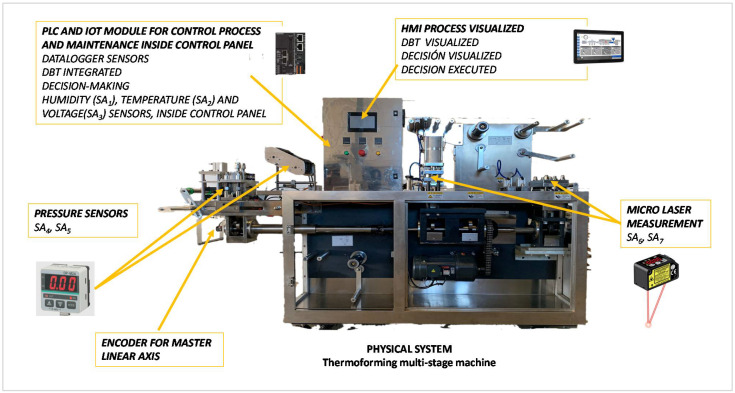
The setup of the DBT maintenance strategy.

**Figure 8 sensors-21-06809-f008:**
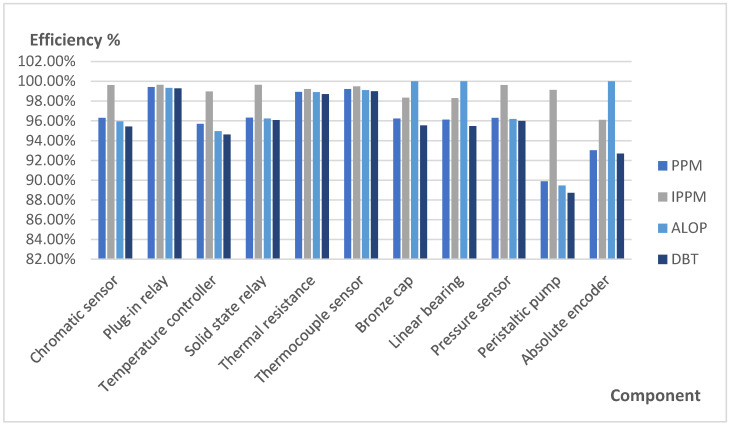
Comparison of efficiency values in affected components.

**Figure 9 sensors-21-06809-f009:**
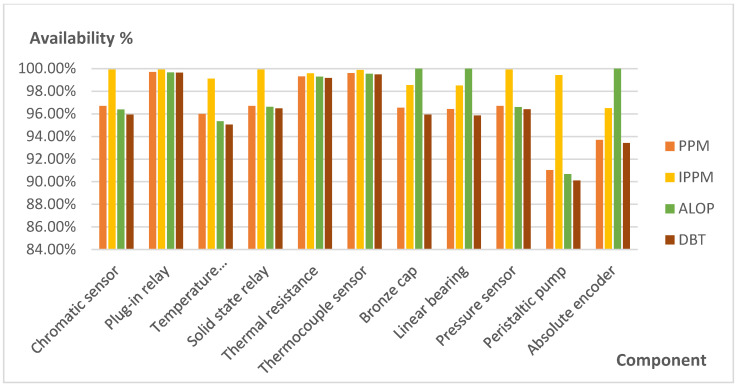
Comparison of availability values in affected components.

**Table 1 sensors-21-06809-t001:** Basic decomposition of components and faults in a multi-stage thermoforming machine.

Item	Component	Type	Fault Source	Consequence of Failure
1	Master power switch	Power Machine/Static	Ambient condition, Power supplier event	Stop
2	PLC	Control/Static	Ambient condition, Power supplier event	Stop
3	HMI	Control/Static	Ambient condition, Power supplier event, Crash	Stop
4	Chromatic sensor	Sensor/Static	Ambient condition, Power supplier event, Crash	Malfunction
5	Plug-in relay	Control device/Static	Ambient condition, Power supplier event	Stop
6	Command and signalling	Control	Ambient condition, Power supplier event, Crash	Stop
7	Safety limit switch	Security/Static	Ambient condition, Power supplier event	Stop
8	Safety relay	Security/Static	Ambient condition, Power supplier event	Stop
9	Safety button	Security/Static	Ambient condition, Power supplier event, Crash	Stop
10	Temperature controller	Control/Static	Ambient condition, Power supplier event	Stop
11	Solid state relay	Actuator/Static	Ambient condition, Power supplier event	Malfunction
12	Thermal resistance	Actuator/Dynamic	Global fatigue	Malfunction
13	Thermocouple sensor	Control/Dynamic	Global fatigue	Malfunction
14	Tape drive	Actuator/Static	Ambient condition, Power supplier event	Stop
15	Tape Motor	Motor/Dynamic	Global fatigue	Malfunction
16	Bronze cap	Mechanism/Dynamic	Global fatigue	Malfunction
17	Linear axis	Mechanism/Dynamic	Global fatigue	Malfunction
18	Lineal bearing	Mechanism/Dynamic	Global fatigue	Malfunction
19	Pneumatic valve	Actuator/Dynamic	Pressure failure, Failure valve	Malfunction
20	Pneumatic cylinder	Actuator/Dynamic	Pressure failure, Cylinder failure	Malfunction
21	Pressure sensor	Control/Static	Ambient condition, Power supplier event	Stop
22	Servo drive peristaltic pump	Actuator/Dynamic	Ambient condition, Power supplier event	Stop
23	Peristaltic pump	Actuator/Dynamic	Global fatigue	Malfunction
24	Terrine cutter	Mechanism/Dynamic	Global fatigue, Mechanical hit	Malfunction
25	Absolute encoder	Control/Dynamic	Ambient condition, Power supplier event. Mechanical hit	Stop

**Table 2 sensors-21-06809-t002:** Thermoforming components times in seconds. Efficiency and availability in %.

Component	MTTR	TTRP	TTC	TTMA	TTPR	MTTF	TLP	Efficiency	MTBF	Availability
Master power switch	14,400	3600	0	0	10,800	9,999,999	28,800	99.71%	10,014,399	99.86%
PLC	435,600	3600	86,400	0	345,600	9,999,999	450,000	95.69%	10,4435,599	95.99%
HMI	435,600	3600	86,400	0	345,600	9,999,999	450,000	95.69%	10,435,599	95.99%
Chromatic sensor	176,520	3600	120	0	172,800	5,000,000	190,920	96.31%	5,176,520	96.70%
Plug-in relay	14,400	3600	0	0	10,800	5,000,000	28,800	99.43%	5,014,400	99.71%
Command and signalling	14,400	3600	0	0	10,800	5,000,000	28,800	99.43%	5,014,400	99.71%
Safety limit switch	14,400	3600	0	0	10,800	9,999,999	28,800	99.71%	10,014,399	99.86%
Safety relay	14,400	3600	0	0	10,800	9,999,999	28,800	99.71%	10,014,399	99.86%
Safety button	14,400	3600	0	0	10,800	9,999,999	28,800	99.71%	10,014,399	99.86%
Temperature controller	435,600	3600	86,400	0	345,600	9,999,999	450,000	95.69%	10,435,599	95.99%
Solid state relay	176,400	3600	0	0	172,800	5,000,000	190,800	96.31%	5,176,400	96.70%
Thermal resistance	25,500	14,400	0	300	10,800	3,700,800	39,900	98.93%	3,726,300	99.32%
Thermocouple sensor	14,700	3600	0	300	10,800	3,700,800	29,100	99.22%	3,715,500	99.61%
Tape drive	435,600	3600	86,400	0	345,600	9,999,999	450,000	95.69%	10,435,599	95.99%
Tape motor	187,200	14,400	0	0	172,800	5,000,000	201,600	96.11%	5,187,200	96.52%
Bronze cap	288,000	28,800	0	86,400	172,800	7,750,000	302,400	96.24%	8,038,000	96.54%
Linear axis	288,000	28,800	0	86,400	172,800	7,625,000	302,400	96.18%	7,913,000	96.49%
Lineal bearing	288,000	28,800	0	86,400	172,800	7,500,000	302,400	96.12%	7,788,000	96.43%
Pneumatic valve	176,400	3600	0	0	172,800	9,999,999	190,800	98.13%	10,176,399	98.30%
Pneumatic cylinder	176,400	3600	0	0	172,800	9,999,999	190,800	98.13%	10,176,399	98.30%
Pressure sensor	176,700	3600	300	0	172,800	5,000,000	191,100	96.31%	5,176,700	96.70%
Servo drive peristaltic pump	435,600	3600	86,400	0	345,600	9,999,999	450,000	95.69%	10,435,599	95.99%
Peristaltic pump	547,200	14,400	0	14,400	518,400	5,000,000	561,600	89.88%	5,547,200	91.02%
Terrine cutter	288,000	28,800	0	86,400	172,800	9,999,999	302,400	97.06%	10,287,999	97.28%
Absolute encoder	360,000	14,400	86,400	86,400	172,800	5,000,000	374,400	93.01%	5,360,000	93.71%

**Table 3 sensors-21-06809-t003:** Comparison of efficiency and availability between PPM and IPPM.

Item	Component	PPM		IPPM		Difference IPPM-PPM	
		Efficiency	Availability	Efficiency	Availability	Efficiency	Availability
1	Master power switch	99.71%	99.86%	99.82%	99.96%	0.10%	0.10%
2	PLC	95.69%	95.99%	98.96%	99.11%	3.27%	3.12%
3	HMI	95.69%	95.99%	98.96%	99.11%	3.27%	3.12%
4	Chromatic sensor	96.31%	96.70%	99.63%	99.92%	3.32%	3.22%
5	Plug-in relay	99.43%	99.71%	99.63%	99.92%	0.21%	0.21%
6	Command and signalling	99.43%	99.71%	99.63%	99.92%	0.21%	0.21%
7	Safety limit switch	99.71%	99.86%	99.82%	99.96%	0.10%	0.10%
8	Safety relay	99.71%	99.86%	99.82%	99.96%	0.10%	0.10%
9	Safety button	99.71%	99.86%	99.82%	99.96%	0.10%	0.10%
10	Temperature controller	95.69%	95.99%	98.96%	99.11%	3.27%	3.12%
11	Solid state relay	96.31%	96.70%	99.63%	99.92%	3.32%	3.22%
12	Thermal resistance	98.93%	99.32%	99.21%	99.60%	0.28%	0.28%
13	Thermocouple sensor	99.22%	99.61%	99.50%	99.89%	0.28%	0.28%
14	Tape drive	95.69%	95.99%	98.96%	99.11%	3.27%	3.12%
15	Tape Motor	96.11%	96.52%	99.42%	99.71%	3.31%	3.19%
16	Bronze cap	96.24%	96.54%	98.35%	98.55%	2.11%	2.01%
17	Linear axis	96.18%	96.49%	98.32%	98.53%	2.14%	2.04%
18	Linear bearing	96.12%	96.43%	98.29%	98.51%	2.18%	2.07%
19	Pneumatic valve	98.13%	98.30%	99.82%	99.96%	1.69%	1.66%
20	Pneumatic cylinder	98.13%	98.30%	99.82%	99.96%	1.69%	1.66%
21	Pressure sensor	96.31%	96.70%	99.63%	99.92%	3.32%	3.22%
22	Servo drive peristaltic pump	95.69%	95.99%	98.96%	99.11%	3.27%	3.12%
**23**	**Peristaltic pump**	**89.88%**	**91.02%**	**99.14%**	**99.42%**	**9.26%**	**8.40%**
24	Terrine cutter	97.06%	97.28%	98.72%	98.87%	1.66%	1.59%
25	Absolute encoder	96.12%	96.43%	98.29%	98.51%	2.18%	2.07%

**Table 4 sensors-21-06809-t004:** Sensors and components used for the ALOP model.

Sensor	Description	Items Affected
SA_1_	% humidity inside the control panel	1, 2, 3, 4, 5, 8, 10, 11, 14, 22, 24, 25
SA_2_	Cª temperature inside control panel	1, 2, 3, 4, 5, 8, 10, 11, 14, 22, 24, 25
SA_3_	Voltage RMS in IGBT	1, 2, 3, 4, 5, 8, 10, 12, 14, 15, 19, 20, 21, 22, 23, 25
SA_4_	Pressure sensor for thermoformer tub MODEL DPM2A of PANASONIC	10, 12, 13, 16, 18, 19, 20, 21
SA_5_	Pressure sensor for peristaltic pumps MODEL DPM2A of PANASONIC	22, 23
SA_6_	Micro laser measurement, side front MODEL HGC of PANASONIC	14, 15, 16, 17, 18
SA_7_	Micro laser measurement, side rear MODEL HGC of PANASONIC	14, 15, 16, 17, 18

**Table 5 sensors-21-06809-t005:** Normal pattern of behaviour of multi-stage thermoforming machine.

EP (Encoder Position)	0	10	100	200	300	400	450	500	600	700	800	900	970	980	990	999
AC_1_: Cam bottom dead centre	1	1	1	1	1	0	0	0	1	1	1	1	1	1	1	1
AC_2_: Drag start point	0	1	0	0	0	0	0	0	0	0	0	0	0	0	0	0
AC_3_: Blown Time	0	0	1	0	0	0	0	0	0	0	0	0	0	0	0	0
AC_4_: Start of heater operation	0	0	0	1	1	1	1	1	1	1	1	1	1	1	1	0
AC_5_: Dosing point	0	0	0	0	1	0	0	0	0	0	0	0	0	0	0	0
AC_6_: Top point cams	0	0	0	0	1	1	1	1	0	0	0	0	0	0	0	0
AC_7_: Home pushers	0	0	0	0	0	0	1	1	1	1	1	1	0	0	0	0
AC_8_: Start blowing	0	0	0	0	0	0	1	0	0	0	0	0	0	0	0	0
SA_1_: Humidity % inside Control Panel	60	60	60	60	60	60	60	60	60	60	60	60	60	60	60	60
SA_2_: Temperature inside Control Panel	40	40	40	40	40	40	40	40	40	40	40	40	40	40	40	40
SA_3_: Voltage supplier inside Control Panel	230	230	230	230	230	230	230	230	230	230	230	230	230	230	230	230
SA_4_: Pressure sensor Thermoforming step (bar)	0	0	4	0	0	0	0	0	0	0	0	0	0	0	0	0
SA_5_: Pressure sensor after Peristaltic Pump (bar)	0	0	0	0	0.2	0	0	0	0	0	0	0	0	0	0	0
SA_6_: Laser measure hp heat seal front (Data in mm)	0	0	0	0	5.0	5.0	5.0	5.0	0.0	0	0	0	0	0	0	0
SA_7_: Laser measure hp heat seal rear (Data in mm)	0	0	0	0	5.0	5.0	5.0	5.0	0.0	0	0	0	0	0	0	0

**Table 6 sensors-21-06809-t006:** Comparison of unexpected failures detected for ALOP and DBT.

Item	Component	ALOP True	ALOP False	DBT True
1	Master power switch			
2	PLC			
3	HMI			
4	Chromatic sensor	1		1
5	Plug-in relay	1	1	1
6	Command and signalling			
7	Safety limit switch			
8	Safety relay			
9	Safety button			
10	Temperature controller	1	1	1
11	Solid state relay	1		1
12	Thermal resistance	1	1	1
13	Thermocouple sensor	1	1	1
14	Tape drive			
15	Tape Motor			
16	Bronze cap			1
17	Linear axis			
18	Linear bearing			1
19	Pneumatic valve			
20	Pneumatic cylinder			
21	Pressure sensor	1		1
22	Servo drive peristaltic pump			
23	Peristaltic pump	1		1
25	Absolute encoder			1

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
