# Peer review of "Maintenance Strategies for Industrial Multi-Stage Machines: The Study of a Thermoforming Machine"

_sensors, 2021, doi:10.3390/s21206809_

Round 1

Reviewer 1 Report

Dear authors,

Overall impression:

The article is very interesting but there are many flaws in terms of a thorough preparation.

The first section is very broadly researched, but it needs more visible clarity. The case study in section two needs to be detailed further. In section three more clarity needs to be added, e.g.: optical clarity needs to be established, figures need to be improved and explained in more detail, sources need to be added and argumentation need to be improved. In section 4 it should be concluded having the specific character of “multi-stage” in mind. There should be added a comment on further research”.

Detailed remarks:

For the comment numbers (Comment No) please see attached PDF. Please comment on each comment number.

Comment No

Comment

1

Language expression

2

"Conventional" not clear

3

Not clear, what "first case" is

4

Source(s) missing

5

Is there a logical relation between "single-stage" and "first case"?

6

Definition multi-stage unclear

7

Example for multi-stage missing

8

Orthographical mistake

9

Language expression

10

Language expression

11

"Alternative" not clear. Relation to "multi-stage"?

12

Refer to later definition of MTTR

13

Language expression

14

"Based on the above" not clear

15

"Desing S3-RF" not clear. Typo?

16

Language expression

17

Layout change: DBT is styled like a sub title, but all the other approaches weren't styled like "DBT" -> create subchapters with enumerations?

18

Blank space

19

Strive for a clear optical structure: the maintenance strategies should be associated clearly with paragraphs

20

Language expression

21

Logical link: machine learning <-> security?

22

Language expression

23

"based on computational data" not clear

24

"behavioural models" not clear

25

"the real model" not clear

26

Source [37] is not related to multi-stage

27

EDT not explained in details

28

Layout change: DBT is styled like a sub title, but all the other approaches weren't styled like "DBT" -> create subchapters with enumerations?

29

Section 1 should be structured more clearly e.g. with subchapters

30

The paragraph should be shifted to the section 2 where the setup is explained

31

The specific character of "multi-stage" must be detailed and worked out more intensively

32

Language expression

33

Missing labels in Figure 1 / Bullet points must be found also in Figure 1

34

Missing scale in Figure 1

35

"4.3-7-10" not nice

36

Missing consistency: ";" <-> ","

37

Missing consistency: ";" <-> ","

38

Language expression

39

PLC: all abbreviations must be explained at first usage

40

Missing consistency: Origin <-> Source

41

Consistency/Not clear: consequence of its failure <-> "Failure" in Table 1

42

Consistency: usage of capital letter "Static"<->"static"

43

Consistency: usage of blank space bevor "/"

44

Orthographical mistake

45

Consistency: usage of "."<->","

46

Not clear: "are one of many multi-stage machines"

47

Put step numbers in flag symbols

48

Consistency: "." used

49

Language expression

50

Shaft to be identified in Figure 1?

51

Language expression

52

Language expression

53

Unclear: favourable resolution

54

Source(s) missing

55

Consistency: writing of Multi Stage Thermoforming Machine

56

"the fixed value of MTTF" not clear

57

Unclear: "several strategies" - more than 1 - why?

58

Language expression

59

Language expression

60

Not clear: "market values"

61

-

62

Source(s) missing

63

Not clear what "has its own time values" means. Is this related to line 221ff?

64

Language expression

65

Source(s) missing

66

Top border line of Figure 3 missing

67

"i" should be indexed in Figure 3

68

Consistency: usage of capital letter "Motor"

69

Consistency: usage of capital letter "Cutter"

70

See 55

71

"on the supply of components" is not clear

72

"substituting the supply time …" is not clear

73

Main message of Figure 3 is the difference between IPPM and PPM - this must be pointed out in more details

74

Consistency: usage of capital letter "Efficiency"/"Availability"

75

Highlight "Peristaltic pump" if this is really the only mentionable point of Table 3

76

Why pointing out only "Peristaltic pump" in Text for Table 3

77

Orthographical mistake

78

Orthographical mistake

79

see 68

80

see 69

81

"electronic components" and "mechanical components" from text are not found in Figure 4

82

Language expression

83

Language expression

84

Language expression

85

Language expression

86

Language expression

87

Difference between "expected time" and "known from experience"?

88

Orthographical mistake

89

Language expression: expression = equation?

90

Please improve figure 5

91

Where SA1…SA6 to be found in Figure 5?

92

Language expression: Setup instead of conceptualisation?

93

Consistency: usage of capital letter "Humidity"

94

Consistency: usage of capital letter "Thermoformer"

95

Not defined: WIKA

96

Not defined: HGC

97

Consistency: usage of "items" in Table 1<->"Components" in Table 4

98

Unclear: The adoption of this …"

99

Language expression

100

See 89

101

Not clear: "command and signalling"

102

Brackets around "7 and 8"?

103

"do not apply for this model" - why?

104

"Where MTBFi is the …": not a full sentence

105

MTTRi: please use index for "i"

106

100 machine cycles: why?

107

Language expression

108

Difference R(t,i) in (9) and R_((t,i)) in text?

109

What is Ofi_i?

110

Language expression

111

Language expression

112

Blank space

113

"its shutdown" not clear - language expression?

114

"is evaluated": language expression

115

"x" is for cross product in (10)

116

Source(s) missing for "c" is 0.67

117

"may have" not clear

118

Blank space

119

"was used for all "j" sensors": why?

120

SA1…SAn -> use index: SA1

121

San. -> why "."?

122

Avoid these few words in new line (STEP2, STEP6, STEP8)

123

Blank space

124

Blank space

125

costi -> blank before index in (11)

126

Language expression

127

Language expression

128

"that very moment": not clear

129

"coordinated manner": not clear

130

Language expression

131

proved to be efficient

132

Why Item with capital letter? Consistency with item/component?

133

Language expression

134

See 90

135

Setup instead of conceptualisation?

136

Language expression

137

Language expression

138

Language expression

139

ACZ not defined, Z is index?

140

Use Index for Sai

141

Consistency: usage of capital letter "behaviour map"<->"pattern of behaviour"

142

The Study has been developed by evaluating…: not clear

143

EP / Acz

144

AC.1 to AC.8: Why "."?

145

Not clear: using the encoder assessment scale

146

In short…: not clear

147

Language expression

148

Language expression

149

Any learning: not clear

150

"Proposed DBT" on next page

151

"EP1000)." not nice in new line

152

See 120

153

Use symbol for arrows

154

Blank space

155

Font in equations is italic style, why - (14)?

156

Language expression: "is other than" - "is different from"

157

Language expression: "as to whether"

158

Language expression: "this is"

159

Language expression: reflect

160

measurement scale has been assessed: not clear

161

Language expression: only thing

162

…has its own relationship. Proof?

163

See 97

164

Language expression: without fail

165

failure?

166

See 120

167

"Very precise": not clear

168

See 55

169

Is there a column "DBT false" not visible?

170

All the "1" in the table are not vertically aligned

171

strategies have been tested: test conditions?
"Unexpected failures can be detected with ALOP and DBT": where can i see that in the table?

172

Consistency with item/component?

173

See 120

174

Typo: SA2 and SA3 with index or not, see 120 also

175

See 120

176

Language expression: expression = equation?

177

Top border line of Figure 7 missing

178

Top border line of Figure 8 missing

179

"Comparison" - "Comparative values": consistency

180

What shall i see exactly in the two figures 7 & 8? Please describe in detail in text

181

of Availability values?

182

significance need to be proven. It is a statistical term.

183

Not clear: Given the results

184

Language expression: is interesting

185

Language expression: "could be an avenue"

186

See 120

187

Language expression: to do so

188

dmax / dmin: use index

189

Please come back to the specific character of "multi-stage" in this conclusion

190

What is the further research?

191

Consistency: second name in bibliographic data abbreviated?

192

Blank space

193

Full or abbreviated first names in bibliographic data?

194

Blank space

Best regards,

Reviewer

Author Response

We have just finished the review of our paper (Manuscript ID: sensors-1399311) titled “Maintenance Strategies for Industrial Multi-Stage Machines: The study of a Thermoforming Machine”. We sincerely appreciate the suggestions of the reviewers. We have tried to comply all the comments, which have been used to improve the quality of the original manuscript.

Reviewer 2 Report

The authors propose two new predictive maintenance strategies that allow for the dynamic calculation of mean-time-to-fail (MTTF) values by logging data from strategically placed sensors in order to increase efficiency and availability. The measurements are collected either in predefined intervals or during specific machine phases which are measured by a position sensing device (encoder).

The strategies are then applied to an existing industrial multi-stage thermoforming machine in order to measure efficiency and availability and to compare against existing preventative maintenance strategies that are based on static MTTF values.

The evaluation of the algorithms on an actual machine and the collection of real-world data is very desirable. However, since the data points are limited (due to the infrequent occurrence of actual failures) it is questioned whether they are sufficient to formulate conclusive results.

The main strong point of the paper are:

a) Novel approach by using analog and digital sensors to improve predictive maintenance

b) The idea presented by the authors is both important and practical

c) The paper is overall well structured

With respect to weaknesses,

a) Some comparison points in the conclusion section are not clear and need to be rephrased

b) More datapoints would be useful in order to draw concrete conclusions

c) The literature cited can be improved with more recent publications. For example, with respect to studies based on machine learning which make it possible to work securely and prevent threats in the cloud, the following article could be relevant. 

Α. Short, HC. Leligou, E. Theocharis, “Execution of a Federated Learning process within a smart contract”, IEEE International Conference on Consumer Electronics, 2021, January 10-12, DOI: 10.1109/ICCE50685.2021.9427734

Author Response

(The authors gave the same response as above.)

Reviewer 3 Report

This paper compares the results of diverse maintenance strategies for multi-stage industrial manufacturing machines. The authors analyze a real case of a multi-stage thermoforming machine. Two methods based on preventive maintenance, Preventive Programming Maintenance (PPM) and Improve Preventive Programming Maintenance (IPPM), are compared with two new strategies based on predictive maintenance, namely Algorithm life optimization Programming (ALOP) and Digital Behaviour Twin (DBT). The authors need to state their goals with the resulting experiments clearly. This study may raise the interest of the research community dealing with industrial maintenance.

The manuscript's structure could be improved by introducing a new section to analyze the Related Work, making it clearer the scientific goal of the study. It would be helpful to learn the positive and negative aspects of the commented related work. The authors did that somehow, but I think it may be improved. A table containing a comparison would increase the quality of the review. The description of the adopted method is not clear enough. It is expected to receive details that make the research reproducible. In fact, a methodology section would be helpful before the case study section. The Results and Conclusion section also should contain enough information for leading other groups to reproduce the proposed approach. The discussion of the results is concise and well written, but the conclusion section should be improved before publication.

Author Response

(The authors gave the same response as above.)

Round 2

Reviewer 1 Report

Dear authors,

thank you for the accurate revision of the initial article.

The following points were noticed in the revised document:

  • L155 Typo „researche“
  • L647 „for many“: Language expression
  • L655 „Due to“: Language expression, “because”?
  • L666 Index on dmax and dmin
  • L674 Typo: increase
  • L679 Sai -> index “i”
  • L686 Typo „De“

Best regards,

Reviewer

Author Response

(The authors gave the same response as above.)
